# An Engineered Protein-Based Building Block (Albumin Methacryloyl) for Fabrication of a 3D In Vitro Cryogel Model

**DOI:** 10.3390/gels8070404

**Published:** 2022-06-25

**Authors:** Xueming Niu, Mian Lin, Bae Hoon Lee

**Affiliations:** 1Wenzhou Institute, University of Chinese Academy of Sciences, Wenzhou 325011, China; xueming.niu@uq.edu.au (X.N.); linmian@wiucas.ac.cn (M.L.); 2Oujiang Laboratory (Zhejiang Lab for Rengerative Medicine, Vision and Brain Health), Wenzhou 325000, China

**Keywords:** albumin methacryloyl, cryogels, 3D in vitro models, liver tissue engineering

## Abstract

Drug-induced liver injury (DILI) is a leading cause of attrition in drug development or withdrawal; current animal experiments and traditional 2D cell culture systems fail to precisely predict the liver toxicity of drug candidates. Hence, there is an urgent need for an alternative in vitro model that can mimic the liver microenvironments and accurately detect human-specific drug hepatotoxicity. Here, for the first time we propose the fabrication of an albumin methacryloyl cryogel platform inspired by the liver’s microarchitecture via emulating the mechanical properties and extracellular matrix (ECM) cues of liver. Engineered crosslinkable albumin methacryloyl is used as a protein-based building block for fabrication of albumin cryogel in vitro models that can have potential applications in 3D cell culture and drug screening. In this work, protein modification, cryogelation, and liver ECM coating were employed to engineer highly porous three-dimensional cryogels with high interconnectivity, liver-like stiffness, and liver ECM as artificial liver constructs. The resulting albumin-based cryogel in vitro model provided improved cell–cell and cell–material interactions and consequently displayed excellent liver functional gene expression, being conducive to detection of fialuridine (FIAU) hepatotoxicity.

## 1. Introduction

The liver is a highly vascularized soft organ and performs various vital functions in the human body such as homeostasis, bile acid production, detoxification, and drug metabolism [1]. In the process of drug metabolism, liver is often exposed to the adverse effects of many drugs. Consequently, drug-induced liver injury (DILI) has become a leading cause of drug development termination and withdrawal from the market [2,3,4]. Unfortunately, current preclinical animal testing is unable to precisely detect human-specific drug metabolites and hepatotoxicity because of the difference of the liver enzyme activities in drug metabolism between humans and animals [5,6,7].

Therefore, predictive models for drug hepatotoxicity including 2-dimensional (2D) [8,9] and 3-dimensional (3D) [10,11] in vitro systems have been developed to investigate the potential liver toxicity of drug candidates. Monolayer 2D culture systems using immortalized cell lines and primary hepatocytes have been widely used for liver toxicity testing [8]. However, 2D culture systems cannot recapitulate the liver microenvironments that are strongly correlated with hepatic cellular functions; hepatocytes in monolayer 2D culture systems lose viability rapidly and decrease liver-specific functionality within a few days [11,12,13]. A sandwich-culture system of culturing primary hepatocytes between two layers of collagen improved cell viability, polarized architecture, and liver-specific functions [10]. However, all 2D systems still have limited cell–cell interactions and cellular functions, resulting in inconsistent drug responsiveness.

The limitations of 2D culture systems have driven the development of 3D in vitro culture systems that are categorized into nonscaffold-based systems such as spheroid/cell aggregate culture [14,15] and hanging drop [16,17] and scaffold-based systems such as microfluidic liver models [18], co-culture [19], and 3D printing [11,20] to mimic the complex features of the liver such as enhanced cell–cell communications, multicellular environments, and efficient supply of nutrients and oxygen. Especially, spheroid culture systems can provide excellent cell–cell interactions in the multicellular structure [21], but the formation of necrosis in the center of large spheroids needs to be overcome. The hanging drop method is an efficient tool for generating cell spheroids/aggregates. However, it has a handling issue; the cell aggregates are easily destroyed [22]. Co-culture systems using parenchymal hepatocytes and nonparenchymal cells (endothelial cells, hepatic stellate cells, and Kupffer cells) together were found to maintain the viability and functions of hepatocytes for a long time [23]. Moreover, microfluidic devices can offer round cell aggregates with an in vivo hepatocyte-like morphology and sustain liver-like cell functionality, but they cannot support long-term culture and are not suitable for high throughput applications [7,8,24,25]. Bioprinting in vitro models has been used in drug screening, but exhibited some drawbacks including a complicated procedure, high cost, and a limitation of it being difficult to print a microlevel complex pore structure [20,26]. Hence, developing a robust in vitro model that can emulate the liver microenvironments and precisely predict human-specific liver toxicity still remains a major challenge for the liver tissue engineering and pharmaceutical industry.

Hydrogels as another 3D culture system have been used in a variety of medical applications and tissue regeneration because the hydrogel network exhibits excellent properties similar to natural tissues [27,28]. On the other hand, hydrogels can be designed with various properties to provide an optimal culture microenvironment to cells [22]. However, bulk hydrogel-based cell culture may be hampered owing to a slow exchange of nutrients and oxygen and limit cell movement when bulk hydrogels with a nano mesh size are employed [29]. Therefore, some methods of making interconnected porous hydrogels have been reported [27]. For example, Soon Seng Ng and co-workers [30] fabricated a 3-dimensional hexagonally arrayed lobular human liver tissue by using inverted colloidal crystals (ICCs) made of poly(ethylene glycol)diacrylate (PEGDA). The engineered liver tissues exhibited a controllable uniform pore size and important features of human-specific liver, and accurately predicted the human-specific hepatotoxicity led by fialuridine (FIAU) that escaped hepatotoxic detection in preclinical animal tests. However, the drawbacks of the ICC method include the complicated procedure, relatively high cost, and use of toxic chemicals in its preparative process.

Cryogels are another type of macroporous hydrogels formed at sub-zero temperatures through a process known as cryogelation that is a simple, inexpensive, and eco-friendly method [31]. Cryogel technologies are expected to address challenges in tissue engineering and other emerging bioengineering disciplines. Recently, Boulais et al. has reported an alginate-based cryogel-integrated biochip for liver tissue engineering and drug screening in which cell-laden cryogels with 1.5 kPa (as a healthy liver model) and 29 kPa (as a cirrhotic liver model) were examined for albumin synthesis and glucose consumption for up to 6 days [32]. However, the cryogel chip supported only short-term culture and was not tested for predictive toxicology and drug screening.

Over the years, albumin has been reported to be used for various bioapplications such as wound healing [33] and drug delivery [34,35] owing to its wide availability, biocompatibility, and biological significance [36,37]. However, there are few reports on albumin-based hydrogels for liver tissue engineering and drug screening [37]. Our aim is to engineer albumin-based cryogels for 3D culture and drug screening, and evaluate them in terms of their mechanical properties, interconnected pore structure, cell viability, cell functionality, and human-specific drug (FIAU) hepatotoxicity. To our knowledge, this is the first report on the formation of albumin methacryloyl (a protein-based building block) cryogels as in vitro models for liver tissue engineering and drug screening.

## 2. Results and Discussion

### 2.1. Preparation and Characterization of BSAMA Cryogels with Different Concentrations

The fabrication of an engineered albumin-based liver scaffold for applications in drug screening is illustrated in Figure 1. Under controlled pH and temperature, crosslinkable bovine serum albumin methacryloyl (BSAMA) was obtained via reaction of BSA with methacrylic anhydride (MAA) as seen in Appendix A. NMR spectra of BSA and BSAMA confirmed the successful methacryloylation of BSA to BSAMA (DS: 93.09 ± 0.01%, Appendix A). In comparison with the BSA NMR spectrum, BSAMA NMR contains specific peaks at around 5.4 and 5.6 ppm indicating the acrylic protons (2H) of BSA methacrylamide (CH_2_=C(CH_3_)CONH−). In addition, the methyl protons (3H) of BSA methacrylamide (CH_2_=C(CH_3_)CONH−) appear at around 1.9 ppm. On the other hand, the peak at around 3.0 ppm corresponds to the methylene protons (2H) of unreacted lysine (NH_2_CH_2_CH_2_CH_2_CH_2_-) in BSA. The methylene peak is shifted in the spectrum of BSAMA from around 3.0 ppm to around 3.3 ppm, suggesting the nearly complete reaction of MAA with the free amino groups of BSA. To examine the secondary structure of BSAMA after the methacryloyl of BSA, the circular dichroism (CD) experiments of native BSA and BSAMA were carried out. As shown in Appendix A, the CD spectrum of native BSA displays two negative bands at 209 and 222 nm, which are attributed to α-helical structure; a positive peak appearing at 195 nm is ascribed to β-sheet [36]. The CD spectrum of BSAMA exhibited a pattern similar to that of native BSA; the chemical modification of BSAMA with methacryloyl groups resulted in a slight reduction in the α-helix and β-sheet content. Based on the CD measurements, the fractional helicity value of BSAMA (35.6 ± 0.017%) is lower than those of native BSA (47.2 ± 0.264%). These results are in agreement with the previous report [38,39].

Porous BSAMA cryogels were fabricated by crosslinking BSAMA polymer chains via cryopolymerization in the presence of ammonium persulfate (APS) and tetramethylethylenediamine (TEMED) at −20 °C. APS was used as a chemical initiator to initiate the polymerization whereas TEMED was used as a catalyst to accelerate the generation of radicals [40]. In the design for the fabrication of a liver-like soft porous scaffold, BSAMA samples were dissolved in deionized (DI) water at three different concentrations (5% (B5), 10% (B10), and 15% (B15)) to tune the stiffness and porosity of cryogels. The recipe for fabrication of albumin-based cryogels is presented in Appendix A. To avoid fast polymerization, a lower amount of APS and TEMED was utilized in a sample with a higher concentration. During the cryogelation phase, the ice crystals of a frozen solution acted as porogens. When the cryogelation was completed, the ice crystals melted, and the porous BSAMA cryogel scaffolds were obtained [40,41]. The overall morphology of BSAMA cryogels (B5, B10, and B15) was examined via scanning electron microscopy (SEM) and confocal microscopy, as displayed in Figure 2A. The pore sizes of cryogels were measured via ImageJ software as shown in Figure 2B. Among the cryogels (B5, B10, and B15), B5 exhibited the biggest pore size, ranging from 48 to 96 μm. For B10, the pore size was from 17 to 55 μm. On the other hand, the SEM images of B15 illustrated the presence of dense layers with a space of approximately 10 µm. Figure 2C,D show the porosity and swelling ratio of BSAMA cryogels made with different concentrations. B5 exhibited a higher porosity of 86.87 ± 0.53% compared to B10 (59.82 ± 4.30%) and B15 (44.88 ± 3.69%). Meanwhile, the swelling ratios of B5, B10, and B15 cryogels were 1952.78 ± 85.07%, 1755.31 ± 80.07%, and 759.91 ± 18.25%, respectively. Therefore, decreasing the BSAMA concentration increased the fluid uptake of cryogels. Recently, Lilandra Boulais and co-workers fabricated an engineered alginate-based cryogel-integrated biochip for dynamic hepatoma cell line culture. The pore size and mechanical properties of alginate-based cryogels were similar to those of BSAMA-based cryogels, but the swelling ratio of albumin-based cryogels (B5) was much larger than that of the alginate-based cryogels [32]. The cryostructuring via ice crystal formation in the frozen phase could impact the cross-linking/polymerization procedure in the unfrozen liquid microphase, leading to different porous structures [39]. The average pore size of cryogels could be controlled by adjusting the freezing conditions such as cooling rates/temperatures (−20 °C and −80 °C) during cryogelation [42]. In our cryogel system, the cryogelation temperature of only −20 °C was adopted; therefore, further optimization of the pore structure of albumin-based cryogels remains to be investigated via using different freezing conditions.

The scaffold stiffness is one of the liver’s critical microenvironmental cues. Young’s modulus was determined by the slope of the stress–strain curve from 10% to 20% strain.

It has been reported that the healthy liver shows a Young’s modulus of 6.62 ± 1.41 kPa [43]. Figure 3A shows that B5 exhibited spongy-like properties while B10 and B15 were relatively brittle under compressive pressure. B5 cryogels exhibited excellent shape recovery under a compressive force as seen in Appendix A. It is probably because the cryogelation of B10 and B15 with a higher concentration seemed less controllable than that of B5 with a lower concentration under the current cryogelation settings. The cryogelation of a highly concentrated solution should be further improved and optimized by carefully adjusting cooling rates and temperatures. Further, the machinal properties of BSAMA cryogels were characterized using a dynamic mechanical analyzer (Figure 3B–E). B5 exhibited the highest tolerant strain of 94.9 ± 2.4%, followed by B10 at 79.0 ± 8.0%, and B15 at 47.0 ± 3.6%. B5 exhibited the modulus of 5.65 kPa, which could be comparable to the modulus (~6 kPa) of healthy liver [44]. In contrast, the B15 gels had a Young’s modulus of 15.17 kPa that exhibited mechanical properties in the range of a cirrhotic liver tissue. In addition, the cryogels absorbed water very fast, as demonstrated in Appendix A. The fast water absorption could be conducive to 3D cell culture on the cryogels owing to efficient exchanges of nutrients and wastes. Consequently, macroporous compressible 5% BSAMA cryogels possessed physical and mechanical properties comparable to those of the healthy liver.

### 2.2. Cell-Laden BSAMA Cryogels Support Cell Viability and Proliferation

The presence of cells within the ultrastructure of cell-laden BSAMA cryogels (B5, B10, and B15) was identified using SEM (Figure 4A). In B5 and B10, HepG2 cells were distributed in the scaffolds, and they grew together over the culture period, which is the normal cell behavior of the HepG2 cell line [22]. On the other hand, cells in B15 could hardly be identified at day 1 and day 3; cell aggregates were observed at day 7. As compared to B10 and B15, B5 cryogels seemed to support more cell proliferation probably owing to a bigger pore size and higher porosity, which could provide more efficient supply of nutrients and oxygen to cells. B5 cryogels with the biggest pore size, highest porosity, and highest swelling ratio may provide an optimal condition for cell growth among the scaffolds. Therefore, the pore size and porosity of the scaffolds are the vital factors to affect the cell proliferation and migration inside the cryogels.

Furthermore, the cell proliferation of various 3D BSAMA cryogels was quantified by cell counting kit 8 (CCK-8) at different time points, and the results are shown in Figure 4B. In the early stages of culture, cells within B5, B10, and B15 cryogels proliferated slightly from day 1 to day 3. However, a significant increase in cell proliferation of B5, B10, and B15 was observed at day 7. B5 exhibited the highest proliferation rate among the cryogels, which suggests that the highly porous structure of B5 cryogels could enhance cell growth and migration, and facilitate the transport of nutrients and the removal of metabolic wastes, potentially leading to the maintenance of cellular functionality [22]. Owing to liver-like stiffness, high porosity, and excellent cell compatibility, B5 scaffolds were selected for the next experiments.

### 2.3. B5 Cryogel Scaffolds with ECM Protein Coating

The main liver ECM proteins (collagen type I (C) and fibronectin (F)) were subsequently coupled to B5 scaffolds via N-hydroxysuccinimide (NHS)/1-ethyl-3-(3-dimethylaminopropyl)carbodiimide hydrochloride (EDC) chemistry as displayed in Figure 5A. The presence of ECM proteins on the surface of B5-C, B5-F, and B5-CF was confirmed first by FTIR (Figure 5B). In the ECM-coated B5 cryogels, the peak of amide-I bonds increased greatly at 1648 cm^−1^ due to the stretching vibrations of peptide C=O groups, the peak of amide-II bonds increased at around 1529 cm^−1^ due to N-H bonds, and the stretching peak of C-N groups of amide-III appeared at around 1229 cm^−1^. Moreover, a new broad band was observed at 3118–3490 cm^−1^ resulting from a N-H stretching frequency of amine groups (-NH_2_). In the spectrum of unmodified BSAMA cryogels, the peak intensities of amide-I, amide-II, and amide-III were relatively weak, and the peaks related to amine groups were absent, compared to the ECM-coated B5 cryogels (B5-C, B5-F, and B5-CF). In addition, immunostaining results reconfirmed the successful coating of collagen I and fibronectin proteins on the cryogels (collagen type I: green and fibronectin: red) as seen in Figure 5C. B5-CF scaffolds exhibited the homogeneous coating of collagen type I and fibronectin. Moreover, SEM images were used to ensure that the overall pore size and porous structure of the cryogels appeared intact even after the coating. All ECM-coated BSAMA cryogels maintained an interconnected porous structure (Figure 5D).

### 2.4. Dual ECMs (Collagen and Fibronectin)-Coated BSAMA Cryogels Promote Cell Proliferation Most

The 3D cell experiments were carried out to investigate the cell viability and proliferation of BSAMA cryogels and ECM-modified BSAMA cryogels. Cell viability and proliferation were visualized via live (calcein, green fluorescence)/dead assay (EthD-1, red fluorescence), as seen in Figure 6A, illustrating that the porous architecture of the albumin-based cryogel scaffolds has a positive effect on facilitating cell distribution and proliferation. The growth of cells on the 2D culture substrates was obvious from day 1 to day 7; some dead cells were found at day 14 probably because of the limited space for cell growth and spreading. In contrast, some viable cells were observed in a cross-section of 3D scaffolds (B5, B5-C, B5-F, and B5-CF) at day 1; cells within the 3D scaffolds appeared to proliferate over time. At day 7 and day 14, the significant increase in cell density in B5, B5-C, B5-F, and B5-CF scaffolds was observed whereas dead cells were scarce. Moreover, live cells with a higher density appeared visible on the B5-CF scaffolds when compared to cells cultured in B5, B5-C, and B5-F scaffolds.

A quantitative analysis of cell proliferation with the CCK-8 assay is presented in Figure 6B. The cell proliferation rate in 2D culture has a marked increase from day 1 to day 7 (*p* < 0.005), but there was a steep drop at day 14, which is consistent with the results of the live/dead assay on the 2D substrate. That also might account for the limitations of 2D culture as well as the unreliable drug cytotoxicity evaluation on the 2D culture substrates. On the contrary, a consistent increase in cell proliferation was observed in the 3D albumin cryogel scaffolds. Especially, dual ECM-coated cryogels (B5-CF) exhibited a significant advantage of cell proliferation, when compared to bare cryogels (B5) and other single ECM-coated cryogels (B5-C and B5-F). At day 14, the percentage of cell proliferation increased 2.98-fold, 3.39-fold, 3.47-fold, and 3.85-fold for B5, B5-C, B5-F, and B5-CF culture groups, respectively, as compared to the same cryogel type at day 1.

Liver-produced albumin is known to possess different cell surface receptors [45]. It has been reported that albumin-based hydrogels encouraged the proliferation of a wide variety of cell types such as osteoblasts [46] and human-induced pluripotent stem cells (hiPSCs) [47,48]. In addition, the binding ability of serum albumin to hepatocytes has been confirmed [49]. It is presumed that albumin-based scaffolds might support hepatocyte culture to some extent. In liver, the hepatocytes are surrounded mainly by various extracellular matrices (ECMs) in their microenvironments. It has been shown that ECM proteins play an important role in liver organogenesis and the specific dedifferentiation programs of hepatocytes [50]. Especially, collagen type I and fibronectin are the main noncellular components that can influence the development and function of liver cells [50,51,52]. The integrin binding sites in fibronectin and collagen have been confirmed to promote cell adhesion. Therefore, dual ECM (collagen and fibronectin) coating is an optimal method to mimic the liver microenvironment to facilitate cell development and stabilize the cell functions in engineered albumin-based cryogel liver tissues [53,54].

### 2.5. Porous Albumin-Based Cryogels Support Cell Infiltration/Migration

After seeding HepG2 cells into albumin-based cryogel scaffolds (6 mm in diameter and 2.5 mm in thickness), cells were expected to migrate into the cryogels because albumin-based cryogel scaffolds exhibited an interconnected porous structure as presented in Figure 5. Firstly, cell distribution in cryogels was assessed by SEM images of the surface and inside of the scaffolds, as well as confocal microscopic images. When cells were seeded on the top of the porous albumin-based cryogel scaffolds, few cells were found in middle sections (Appendix A). Seven days after seeding cells (Appendix A), more cells were found to spread on the surface and accumulate in the middle of the cryogel scaffolds. At day 14 (Appendix A), a great number of cells were found inside the scaffolds. To further characterize the infiltration of cells into scaffolds, cell nucleus, f-actin, and the specific protein expression (albumin and CYP3A4) inside the cell-laden scaffolds were further examined in the cross-section images of cell-laden scaffolds as seen in Figure 7A–G. After cell seeding, cells were found on the surface of the scaffolds, but few cells were found within the scaffolds. Over time, the deeper infiltration of cells was detected within the scaffolds, and the deepest accumulation of cells was identified at day 14 post seeding. The effect of ECM proteins on cell infiltration in ECM-coated BSAMA cryogels (B5-C, B5-F, and B5-CF) was greater than in the bare BSAMA cryogels (B5). Specifically, the most significant infiltration was observed in B5-CF. The total depth of cryogel scaffolds was about 2.5 mm; around 12% of the scaffold in depth was observed to be used for cell migration and infiltration judging from the confocal results, which suggested that these 3D BSAMA cryogel scaffolds still had more space for further cell growth and migration.

It is suggested that ECM proteins such as fibronectin and collagen type I could promote cell adhesion and migration [55]. The cell infiltration is an important precondition of the further evaluation of liver cell constructs. [56] In sum, all of the porous BSAMA cryogels demonstrated a consistent cell infiltration. Porous bare albumin-based cryogel scaffolds (B5) provided cells with relatively suitable cell migration; more importantly, ECM-coated albumin-based cryogel scaffolds (B5-C, B5-F, and B5-FC) offered a better microenvironment (liver-like ECMs) to cells for cell adhesion and cell infiltration. Comparing with other 3D cultures, such as hepatocyte spheroid, the porous cryogels could provide extra space to transfer nutrition and oxygen, solving intraspheroid necrosis to some extent [22].

### 2.6. Cell-Laden ECM-Coated Albumin-Based Cryogels Promote Hepatic Functions

In order to examine the influence of these porous albumin-based cryogels on cell specific functions, cells on 2D substrates and in 3D albumin-based cryogels were evaluated via immunofluorescent staining and RT-qPCR assay as displayed in Figure 8 and Figure 9. The target hepatic markers of immunofluorescent staining were albumin and CYP3A4 that are an important secretory protein and a drug metabolic enzyme, respectively. The confocal images show some changes in the expression level of intracellular albumin and CYP3A4 as well as in cell morphology during the culture period, on day 1 (Figure 8A,D), day 7 (Figure 8B,E), and day 14 (Figure 8C,F). Overall, cells on the 2D substrates appeared to express a relatively good amount of albumin and CYP3A4 from day 1 to day 7 and seemed to decrease their expression later on day 14. In addition, cells on the 2D substrates seemed to lose their original cell morphologies (clear forms of cell aggregates) at day 14. By contrast, cells in the 3D albumin-based cryogels started to form small aggregates and then became big aggregates over the culture period, potentially leading to improved cell–cell interactions and cellular functions. Notably, an intense expression level of CYP3A4 and albumin especially in B5-CF scaffolds was observed at day 14. Over time in culture, the cell clusters developed themselves into spherical aggregates where it was difficult to distinguish a single cell anymore. The size of the growing hepatic aggregates became bigger over time. Hence, cells on the 3D albumin-based cryogels aggregated together during the culture period, which could improve cell–cell interactions [57].

Furthermore, to examine the cell function at a gene level, hepatocyte-specific gene expression of HepG2 cells on 2D substrates and in 3D cryogel scaffolds was quantified by RT-qPCR, and the results of day 14 are shown in Figure 9. Hepatocyte-specific genes including albumin, alpha 1-antitrypsin (AAT), CYPs (CYP3A4 and CYP3A7), glucose 6-phosphatase (G6Pase), and hepatocyte nuclear factors (HNFs) were used as liver-specific markers. AAT and G6Pase are largely produced in the liver, which play an important role in balancing the action of neutrophil-protease enzymes and providing glucose during starvation, respectively. The HNF gene is involved in regulating the transcription of a diverse group of genes into proteins. E-cadherin, N-cadherin, claudin-1, and zonula occludens (ZO-1) were assessed as the main junction protein genes [58]. E-cadherin and N-cadherin play an essential role in modulating crucial morphogenetic differentiation processes during development [59,60].

In general, every tested gene was highly upregulated in the 3D albumin cryogel scaffolds as compared to those in the 2D substrates. The gene expression level of albumin and CYP3A4 increased more greatly in the 3D albumin-based cryogels than those on the 2D substrates. Notably, the expression level of albumin and CYP3A4 in B5-CF was the highest among the albumin-based cryogel scaffolds, which is consistent with the results of albumin and CYP3A4 immunostaining. Moreover, a similar pattern was observed in the expression of alpha 1-antitrypsin (AAT), hepatocyte nuclear factor (HNF4α), glucose 6-phosphatase (G6Pase), and CYP3A7. Particularly, cells within B5-CF expressed AAT, G6Pase, and HNF4α at the highest levels among the other samples. As for cell junction genes, the secretion level of zonula occludens (ZO)-1 in B5-CF, a component of the junctional complex between cells that can play a vital role in cell–cell direct contacts via functioning in signal transduction pathways related to gene expression and cell behavior [61], was significantly higher than that in the other samples. On the other hand, claudin-1 was highly upregulated in 3D albumin-based cryogel samples, and the expression level of claudin-1 in B5-CF was higher than that of B5 (*p* < 0.01), B5-C (*p* < 0.0001), and B5-F (*p* < 0.005). Finally, cadherins (E-cadherin and N-cadherin) are a type of cell adhesion molecule (CAM) and are important in the formation of adherent junctions to allow cells to adhere to each other. The expression level of E-cadherin and N-cadherin in B5-CF scaffolds was the highest among the samples. E-cadherin is required for contact-induced polarity in many epithelial cell types [60]. It was reported that hepatocytes had a unique polarized microenvironment to exert their particular activities in the 3D environments of the liver. [62] Therefore, it is paramount to maintain hepatic polarization, and the highest E-cadherin expression level in B5-CF indicates that B5-CF could offer cells a liver-mimetic polarized microenvironment to maintain the hepatocyte-specific features. The extracellular matrix (ECM) of the liver contains various proteins (e.g., collagen, fibronectin, and laminin). Our gene expression results demonstrate that the dual ECM (collagen and fibronectin) coating strategy can more closely mimic the ECM environments of liver and produce better hepatic functions than the single ECM coating, which is consistent with other reports [54,63].

### 2.7. Cell-Laden Albumin-Based Cryogels Predict the Hepatotoxicity of Fialuridine (FIAU) in a Dose- and Time-Dependent Manner

Fialuridine (FIAU) is a nucleoside analogue that was examined for the treatment of chronic hepatitis B virus infection and was unfortunately found to be hepatotoxic in the clinical trials: Seven of 15 clinical trial recipients with FIAU administration ended up in an acute liver failure [64]. Five participants died and two needed a liver transplant. The chemical structure of FIAU is shown in Figure 10A. Fialuridine toxicity is probably mediated by its incorporation into the mitochondrial DNA (mtDNA). In addition, internucleotide linkages with the 3′-hydroxyl group of FIAU leads to impaired gene function [65]. When FIAU is incorporated into mtDNA, FIAU can induce the inhibition of the gamma-DNA polymerase, resulting in mitochondrial dysfunction.

Preclinical animal toxicology studies including mice, rats, dogs, and primates failed to indicate the hepatoxicity of FIAU. It has been reported that owing to the expression of human equilibrative nucleoside transporter 1 (hENT1) in the mitochondrial membrane of humans, the entrance of FIAU into the human liver mitochondrial membrane was promoted, leading to hepatotoxicity as well as loss of liver functions [5]. However, the specific expression of nucleoside transporters is absent in rodents, which is why small animal experiments cannot provide an accurate hepatotoxicity evaluation result [5].

For drug toxicity evaluation, B5-CF cryogels were selected as the optimal albumin-based scaffold for 3D hepatocyte culture. As presented in Figure 10A, HepG2 cells in B5-CF were cultured in media alone (the control) or media with the addition of FIAU (50 µM or 500 µM) every two days over a period of 14 days since the 3D BSAMA cryogels supported the functionality of hepatocytes for 2 weeks. HepG2 cells were employed owing to the advantages of availability and robustness of predicting hepatotoxicity of drug compounds with a certain degree of sensitivity and specificity [66,67]. The hepatotoxicity was assessed in terms of cell proliferation, LDH production, albumin secretion, and ATP content. In our experiments, a relatively low dosage of FIAU (50 µM) did not cause a significant change in cell proliferation until day 5 but started to decrease cell proliferation at day 7 and significantly lowered it at day 14 (66.86 ± 11.19%, *p* < 0.05), as compared to the same group at day 1, indicating that FIAU could inhibit the growth of cells (Figure 10B). On the other hand, the cell proliferation at a high dosage of FIAU (500 µM) showed a decreasing trend during the test period; it significantly decreased at day 5 (67.21 ± 9.24%, *p* < 0.05), day 7 (49.02 ± 14.41%, *p* < 0.005), and day 14 (10.14 ± 1.26%, *p* < 0.001). As seen in Figure 10C, the release pattern of LDH produced from dead cells displayed a reverse trend, as compared to cell proliferation. The cytotoxicity of FIAU was also dose- and time-dependent. The level of LDH remained relatively stable in the control group without the FIAU treatment whereas that of LDH in the FIAU treatment had an increasing trend. Cells within B5-CF responded to the higher dosage of FIAU more sensitively than to the lower dosage of FIAU. The high dosage group (500 µM of FIAU) started to exhibit a significant high level of LDH even at day 3 (*p* < 0.01) onwards, and reached the highest LDH level at day 14 (*p* < 0.001). The drug toxicity evaluation of other cryogels (B5, B5-C, and B5-F) is shown Appendix A. They displayed a similar pattern to B5-CF. Overall, the ECM-coated BSAMA cryogels detected the cytotoxicity of FIAU earlier than the bare BSAMA cryogels. In addition, FIAU influenced cell functions such as albumin production and ATP content in a dose-dependent manner. The standard curves of albumin and ATP are shown in Appendix A. The trend of albumin expression was similar to that of cell proliferation (Figure 10D). A relatively low dosage of FIAU (50 µM) decreased the albumin production abruptly (37.71 ± 11.25%, *p* < 0.005) at day 14 whereas a high dosage of FIAU (500 µM) started to lower albumin production even at day 5 and showed the lowest albumin production (5.66 ± 2.09%, *p* < 0.001) at day 14, probably owing to drug-caused cell death. As seen in Figure 10E, the ATP content in the cell-laden B5-CF cryogels also dropped in a dose-dependent manner at day 14. As mentioned previously, hENT1 is expressed in the human mitochondria and can facilitate the entry of FIAU into the mitochondria. In the mitochondria, thymidine kinase (TK2) monophosphorylates FIAU, and further phosphorylation to triphosphorylated FIAU is mediated by thymidylate kinase (TMPK) and nucleoside diphosphate kinase (DPK) [64,65]. FIAU and its metabolites can incorporate into mitochondrial DNA (mtDNA), consequently decrease mtDNA replication, and cause mitochondrial disfunction [68]. Moreover, the mitochondrion is the main organelle in which energy is generated. Specifically, adenosine triphosphate (ATP) is produced in the inner membrane of the mitochondrion. Therefore, the dose-dependent ATP decrease indicates the severe toxicity of FIAU on the mitochondrial functions [30].

The in vitro hepatotoxic effects of FIAU could depend on drug dosage, cell type, cell density, culture period, and culture system (2D or 3D). It was reported that toxic effects of FIAU were found at around a 10 µM concentration in 2 weeks in a 2D HepaRG system [69]. HepaRG cells appeared to be more sensitive and more suitable toward FIAU toxicity prediction than HepG2 cells. In addition, 3D culture models are more conducive to detection of chronic drug hepatotoxicity over 2D culture systems owing to in vivo-like features [54,70]. Overall, our protein-based cryogels made of mainly albumin methacryloyl as a building block featured a highly porous interconnected structure, had liver ECM coating capability, supported 3D cell culture, and exhibited drug screening potential. In a future study, our 3D albumin-based cryogel system will be further investigated using other cell types including HepaRG, human primary hepatocytes, and total liver cells to further prove its drug screening capability.

## 3. Conclusions

In summary, we reported a 3D in vitro model of cell-laden albumin-based cryogels for applications in drug screening for the first time. Albumin-based cryogels with ECM coating could offer physiologically relevant liver microenvironments in terms of liver-like mechanical properties, highly interconnected porous structure, and liver ECM, leading to promotion of cell–cell interactions and cellular functions. Furthermore, the in vitro model of cell-laden albumin-based cryogels played a role as an engineered liver tissue in evaluating the human-specific toxicity of FIAU. However, there is still some future work waiting to be further investigated. For example, albumin-based cryogels should be proven to be suitable for primary hepatocyte culture or co-culture as well as for testing for multiple drugs. In addition, albumin-based cryogel models should be further improved and tested in terms of high throughput. Together, new albumin-based cryogels could have potential of being used as a versatile platform for 3D cell culture, tissue engineering, and drug screening applications.

## 4. Experiments and Methods

### 4.1. Materials

Bovine serum albumin (BSA) and methacrylic anhydride (MAA) were ordered from Sigma-Aldrich (Shanghai, China). Ammonium persulfate (APS) and tetramethylethylenediamine (TEMED) both were purchased from Rhawn (Shanghai, China). Cell counting Kit-8 (CCK-8) and lactate dehydrogenase-cytotoxicity assay kit (LDH) were purchased from Dojindo Molecular Technologies (Fuzhou, China). Collagen type I was purchased from Macklin (Shanghai, China), and fibronectin was purchased from Yeasen (Shanghai, China). Dulbecco’s modified Eagle’s medium (DMEM), fetal bovine serum (FBS), and antibiotic/antimycotic were purchased from Gibco. Live/Dead^®^ Cell Viability/Cytotoxicity kit was purchased from Life Technologies (Shanghai, China). Fialuridine (FIAU) was ordered from Tokyo Chemical Industry (Tokyo, Japan).

### 4.2. Preparation and Characterization of BSAMA Cryogels with Different Concentrations

Bovine serum albumin methacryloyl (BSAMA) was prepared according to previous reports [36,71]. Briefly, BSA (20.00 g) was allowed to react with methacrylic anhydride (MAA, 6.23 mL) under controlled pH (pH 8–9) and temperature (37 °C) for 1–2 h to obtain BSAMA. 2,4,6-trinitrobenzenesulfonic acid (TNBS) assay was used to calculate the degree of methacryloylation (DM). BSA and BSAMA samples were separately dissolved at 2 mg mL^−1^ in 0.1 M sodium bicarbonate buffer (NaHCO_3_, Sigma-Aldrich, St. Louis, MO, USA). Then, 0.5 mL of each sample was allowed to react with 0.5 mL of 0.1% *w*/*v* TNBS (Sigma-Aldrich) for 2 h at 37 °C. After that, 0.25 mL of 1 M HCl and 0.5 mL of 10 *w*/*v*% sodium dodecyl sulfate (SDS, Sigma-Aldrich) were added to terminate the reaction. Finally, the absorbance of each sample was measured at 335 nm using a microplate reader (Thermo Fisher, Waltham, MA, USA). Glycine solutions (64, 32, 16, 8, 4, 2, 1, and 0 μg mL^−1^ in 0.1 M NaHCO_3_) were utilized to prepare a standard curve. The following formula was used to calculate the DM:$$DM\;\left(\%\right)=\left(1-\frac{M_{BSAMA}}{M_{BSA}}\right)\times 100$$
where *M_BSAMA_* and *M_BSA_* are the molarity of the free primary amino groups in the BSAMA and BSA samples, respectively.

For proton nuclear magnetic resonance (^1^H NMR) spectroscopy, BSA and BSAMA samples were prepared at 75 mg mL^−1^ in deuterium oxide (D_2_O; Merck) at 4 °C for 12 h. The chemical shift of BSA and BSAMA was observed by ^1^H NMR spectroscopy on a Bruker Avance-I 500 MHz spectrometer (Weinheim, Germany). To compare the secondary structure of BSA and BSAMA samples, circular dichroism (CD) experiments in the UV spectral range from 260 to 180 nm were conducted using Chirascan Plus (Applied Photo physics, Leatherhead, UK).

BSAMA samples were then used to fabricate cryogel scaffolds at different concentrations. APS and TEMED were used as a free-radical initiator and a catalyst, respectively. In brief, via adjusting the ratios of APS and TEMED, 5%, 10%, and 15% (*w*/*v*) BSAMA solutions for cryogelation were prepared as seen in Appendix A. The BSAMA solutions were moved into a 96-well plate with 200 µL per well, and kept in −20 °C for 24 h. After that, they were thawed at room temperature for 30 min, and were washed with PBS three times to remove unreacted monomers and crosslinkers. Finally, BSAMA cryogels with different concentrations (5%, 10%, and 15% (*w*/*v*)) were stored at 4 °C for further use.

Surface morphology and porous structure of BSAMA cryogels were assessed by scanning electron microscopy (SEM, SU8010, Hitachi Ltd., Tokyo, Japan) after coating freeze-dried BSAMA cryogels with platinum and by confocal laser scanning microscopy (Nikon A1, Tokyo, Japan) after immersing BSAMA cryogels in a rhodamine solution of 2 µg mL^−1^ for 5 min.

A compression test (UTM2102; Shenzhen, China) was conducted to evaluate the mechanical properties of BSAMA cryogels. The compressive strain and stress were recorded while the cryogels with a thickness of 4 mm and a diameter of 6.5 mm were compressed at a crosshead speed of 10 mm min^−1^ until the cryogels were crushed. The compressive modulus was measured as the slope of a strain–stress curve from 10 to 20% strain. Each sample was tested with three replicates.

In addition, a swelling ratio and porosity of BSAMA cryogels were measured via the volume of PBS (pH 7.4) absorbed by BSAMA cryogel scaffolds [32,72].
$$\mathrm{Porosity}\;\left(\%\right)=\left(m_{h}-m_{d}\right)/m_{h}\times 100$$
$$\mathrm{Swelling}\;\mathrm{ratio}\;\left(\%\right)=\left(m_{h}-m_{l}\right)/m_{l}\times 100$$

*m_h_*: the weight of the hydrated BSAMA cryogel samples in PBS overnight at 37 °C,

*m_d_*: the dehydrated weight of BSAMA cryogel samples after wicking on absorbent papers,

*m_l_*: the dried weight of BSAMA cryogel samples after lyophilization.

### 4.3. Preparation of Cell-Laden BSAMA Cryogel Scaffolds

Primary human hepatocytes (PHHs) have been considered as the “gold standard” for drug screening studies [73]. However, a significant difference between different donors can be found in in vitro experiments, and the limited resource of human liver samples, difficult isolation procedure, and short life span hinder the research of PHHs to some extent [15,74]. Therefore, immortalized liver hepatocellular carcinoma cells (HepG2) are widely used as an alternative cell model to primary hepatocytes due to ease of use, low cost, high reproducibility, and wide availability. More importantly, HepG2 cells easily form self-aggregating spheroids expressing PHH critical markers, albumin, and drug metabolism enzymes and maintain the liver principal functions [75]. HepG2 cells were purchased from Kunming Cell Bank and cultivated in DMEM with 10% FBS and 1% antibiotic/antimycotic at 37 °C in a 5% CO_2_ atmosphere. For preparation of sterile BSAM cryogels (6.5 mm in diameter and 2.5 mm in thickness), BSAMA samples were dissolved in deionized (DI) water at three different concentrations (5, 10, and 15% (*w*/*v*)), and then these BSAMA solutions were filtered using sterile 0.2 μm filters. A 100 µL sterile sample solution was then cast in each wall of a 96-tissue culture plate, which was sealed by a sealing film to avoid bacterial contamination and kept at −20 °C for 24 h for cryogelation. After washing the BSAMA cryogels with sterile PBS three times, 20,000 cells in 20 µL were dropped on the top of each cryogel, and cells were allowed to attach on the scaffolds overnight in a cell incubator (37 °C, 5% CO_2_). Then, 500 µL of the culture medium was added to each well to culture the cell-laden BSAMA scaffolds. The 2D substrates (12-well plates) as a control group were cultured with the same number of cells (20,000 cells in 500 µL). Cell viability was assessed on day 1, day 3, and day 7 using the Cell Counting Kit-8 (CCK-8; Dojindo Molecular Technologies, Rockville, MD, USA) assay. Briefly, a CCK-8 solution (serum-free media: CCK-8 = 10:1) was added to each group, and then each group was incubated at 37 °C in 5% CO_2_ for 2 h. Subsequently, each supernatant (100 µL) was collected, and its absorbance at 450 nm was measured. In order to observe cell-laden cryogels, cell-seeded cryogels were fixed using 4% paraformaldehyde for 30 min, and then gently washed twice with a PBS solution three times. Afterwards, the fixed samples were frozen overnight at −80 °C, and lyophilized for 48 h. After the surfaces of the freeze-dried cryogels were coated with silver, the images of cell-laden cryogels were acquired with a SEM at 5 kV.

### 4.4. Preparation and Characterization of ECM Protein-Coated BSAMA Cryogels

Based on the previous experimental results, 5% BSAMA cryogel scaffolds (B5) were selected for further studies. Collagen type I and fibronectin occupying an important position in liver extracellular matrix (ECM) were chosen to modify the surface of B5 cryogels. First, the -COOH group of BSAMA cryogels was activated by 40 mM N-ethyl-N′-(3-dimethylaminopropyl) carbodiimide (EDC) and 80 mM N-hydroxysuccinamide (NHS) in a 2-(N-morpholino) ethanesulfonic acid (MES, 50 mM) buffer solution for 30 min. Then, the activated cryogels were immersed in a collagen I solution (200 µg mL^−^^1^) or/and fibronectin solution (200 µg mL^−1^), followed by shaking at 300 rpm for 24 h (37 °C). Activated carboxyl groups of BSAMA cryogels reacted with the amino groups on ECM proteins, which resulted in B5–collagen I (B5-C), B5–fibronectin (B5-F), and B5–collagen I–fibronectin (B5-CF). Coated BSAMA cryogels were washed with PBS three times and stored at 4 °C for further use.

Attenuated total reflection–Fourier transform infrared (ATR-FTIR) spectra were recorded on an FTIR spectrometer (Tensor II, Bruker, Germany) to validate the presence of ECM proteins on the surface of BSAMA cryogels. Samples were freeze-dried for 24 h, and then scanned over the wavenumber range of 1000–4000 cm^−1^ using a 4 cm^−1^ resolution and an average of 128 scans. Moreover, immunofluorescence microscopy was used to visualize the distribution of collagen type I and fibronectin on the surface of cryogels. Briefly, the ECM-coated BSAMA cryogels were fixed with 4% paraformaldehyde (PFA) for 5 min, and incubated in a 1% (*w*/*v*) bovine serum albumin blocking solution for 1 h at 4 °C. B5-C, B5-F, and B5-CF were successively incubated (4 °C, overnight) with either rabbit anti-collagen type I primary antibody (1:100, Affinity Biosciences, Cincinnati, OH, USA) or mouse anti-fibronectin primary antibody (1:100, Affinity Biosciences). Afterwards, samples were washed with PBS three times to remove unbound primary antibodies. After that, samples were incubated with either goat anti-rabbit secondary antibody conjugated with DyLight 488 (1:50, Earthox, Millbrae, CA, USA) or goat anti-mouse secondary antibody conjugated with DyLight 594 (1:50, Earthox) at room temperature for 2 h. Finally, samples were washed with PBS three times and observed using a confocal laser scanning microscope (Nikon A1, Tokyo, Japan).

### 4.5. Cell Viability

The cell viability of BSAMA cryogel samples (B5, B5-C, B5-F, and B5-CF) was evaluated qualitatively by a Live/Dead Cell Viability/Cytotoxicity Kit according to the manufacturer’s protocol. Briefly, 2 μM calcein-acetomethoxy (calcein-AM) was used to stain live cells as green, and 4 μM ethidium homodimer-1 (EthD-1) was used to stain dead cells as red. After a 1 h incubation at 37 °C with 5% CO_2_, samples were observed with a confocal laser scanning microscope (Nikon A1, Tokyo, Japan). In addition, a Cell Counting Kit-8 (CCK-8; Dojindo Molecular Technologies, Rockville, MD, USA) assay was used to evaluate the cell proliferation of cell-laden BSAMA cryogels at day 1, day 7, and day 14. Additionally, the morphology of cells inside the scaffolds was observed by SEM.

### 4.6. Immunofluorescence Staining

Immunofluorescence staining of intracellular albumin and CYP3A4 was conducted to examine a liver cell-specific function within the BSAMA cryogels. At day 1, day 7, and day 14, cell-laden samples were washed with PBS for three times, and fixed with 4% paraformaldehyde (PFA) for 5 min, and then blocked and permeated with TritonX-100 (0.1% *w*/*v*) in a BSA solution (1% *w*/*v* in PBS) for 1 h at 4 °C. The samples were incubated with either rabbit anti-albumin primary antibody (1:100, Affinity Biosciences) or mouse anti-CYP3A4 primary antibody (1:100, Affinity Biosciences) in a BSA solution (1% *w*/*v*) at 4 °C overnight. After that, these samples were washed with PBS for three times and incubated with either goat anti-rabbit secondary antibody or goat anti-mouse secondary antibody, both conjugated with DyLight 488 (1:50, Earthox), for 2 h at room temperature. Before the images of cross-sections were captured, these samples were split in the middle of the surface vertically. Confocal microscopic images of CYP3A4 and albumin expression were obtained with a confocal laser scanning microscope (Nikon A1, Tokyo, Japan).

### 4.7. Reverse Transcription–Quantitative Real-Time Polymerase Chain Reaction for Liver Cell Functionality

RT-PCR was used to determine the expression of some liver-specific functional and regulatory genes at day 14. After the 14-day culture, total RNA was isolated and separated from DNA and proteins after extraction with a Trizol solution (500 µL in each sample). Each sample in the Trizol solution was centrifuged at 12,000 rpm for 10 min at 4 °C. Afterwards, total RNA remained in the upper aqueous phase while most of the DNA and proteins remained either in the interphase or in the lower organic phase. The amount of total RNA was measured by a DS-11 spectrophotometer (DeNovix, Wilmington, NC, USA). Then, total RNA was reversed to complementary DNA (cDNA) on a thermal cycler (T 100, Bio-Rad, USA). cDNA was amplified using specific primers and TB Green Premix Ex Taq on a real-time PCR detection system (LightCycler 96, Roche, Switzerland). Briefly, real-time PCR was performed at 95 °C for 15 min, and then it was denatured at 95 °C for 5 s, extended at 60 °C for 30 s, and annealed at 72 °C for 15 s, and was cycled 40 times. The melting curve was prepared from 75 °C to 95 °C with a temperature increase of 1 °C every 20 s [76]. The results were analyzed by the 2^−ΔΔCT^ method. The expression level of each gene was normalized to the expression level of glyceraldehyde-3-phosphate dehydrogenase (GAPDH). The specific primer sequences used for cDNA amplification were selected from the literature shown in Appendix A [58].

### 4.8. FIAU-Induced Hepatotoxicity Assays

In order to evaluate the capability of the engineered albumin-based cryogel liver constructs to accurately predict the potential liver cytotoxicity of a compound, fialuridine (FIAU), that has been reported to be associated with drug-induced liver injury (DILI) hepatotoxicity, was used [30]. FIAU was tested at a low dose (50 μM) and a high dose (500 μM). To test the time-dependent toxicity, cell-laden BSAMA cryogels were incubated with FIAU-containing media for 14 days, and media were changed every two days. Cell viability and cytotoxicity were determined using CCK-8 and LDH, respectively. In addition, a hepatic-specific function was evaluated by human albumin ELISA assay (Mskbio, Wuhan, China) in cell culture media according to the manufacturer’s instruction. This kit includes a human albumin-specific antibody to recognize native human albumin (not bovine albumin) in cell culture supernatant samples. Moreover, a mitochondrial function was also assessed via adenosine 5′-triphosphate (ATP) ELISA assay (Beyotime, Shanghai, China) at the end of the FIAU treatment.

### 4.9. Statistical Analysis

GraphPad Prism 6.01 (GraphPad Software) and OriginPro 8.5.1 were utilized to analyze the data unless otherwise specified. Triplicate samples for each condition were used unless otherwise specified, and data of the results were presented as mean ± standard deviation (SD). Comparisons between the two samples were made using a two-tailed paired *t*-test. A one-way ANOVA was performed to test for differences among at least three groups. In statistical comparisons, *p*-values less than 0.05 were considered statistically significant.

## Figures and Tables

**Figure 1 gels-08-00404-f001:**
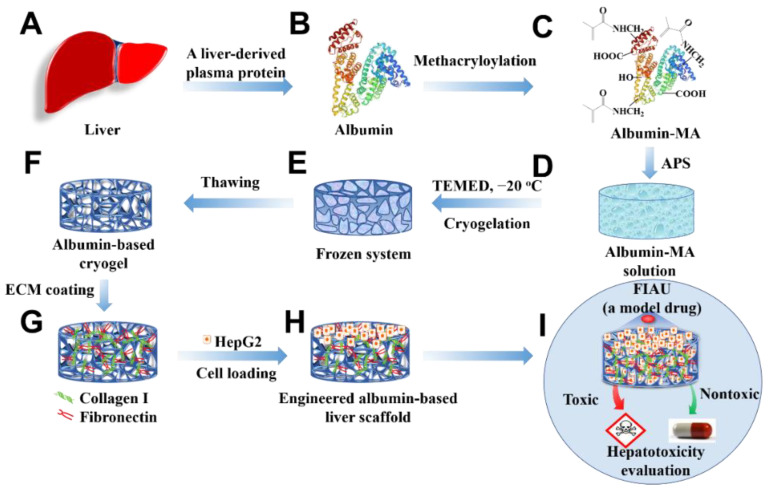
Schematic illustration presenting the fabrication of an engineered albumin-based liver scaffold for hepatotoxicity evaluation. (**A**) Liver. (**B**) Liver-derived albumin. (**C**) Albumin-MA was obtained via the methacryloylation. (**D**) Albumin-MA solution with an initiator, APS. (**E**) Cryogelation with a catalyzer (TEMED) under −20 °C. (**F**) Albumin-based cryogels after thawing. (**G**) ECM-coated albumin-based cryogels. (**H**) Cell-laden albumin-based cryogel liver constructs. (**I**) Hepatotoxicity evaluation.

**Figure 2 gels-08-00404-f002:**
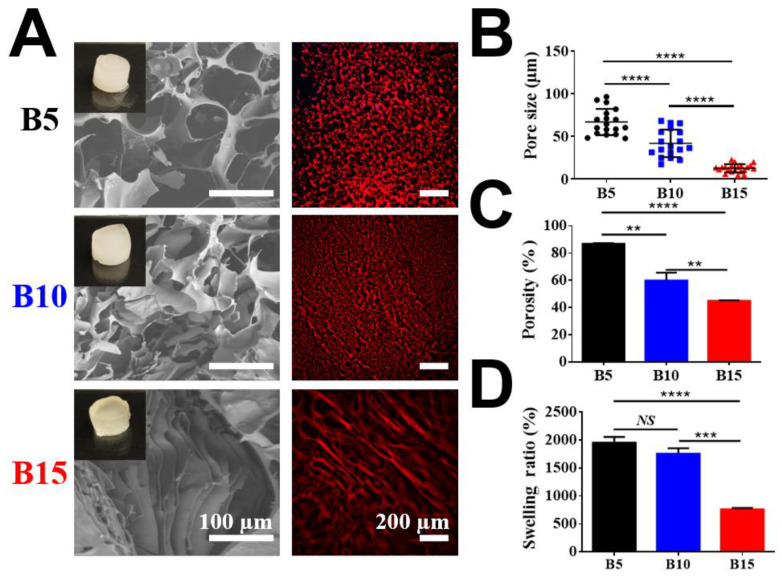
Physical characterization of three BSAMA cryogels (B5, B10, and B15). (**A**) Scanning electron microscopic (SEM) and confocal microscopic images (red: the wall of cryogels). Scale bar: 100 μm in SEM; 200 μm in confocal. (**B**) Pore size distribution. Black circles, blue squares, and red triangles represent the pore sizes of B5, B10, and B15, respectively. (**C**) Pore connectivity was demonstrated as a porosity (%). (**D**) Swelling ratio. *n* = 3, **: *p* < 0.01, ***: *p* < 0.005, ****: *p* < 0.001, NS means no significant difference between the samples at the extremities of the line. 18 pores were counted in every kind of BSAMA cryogels.

**Figure 3 gels-08-00404-f003:**
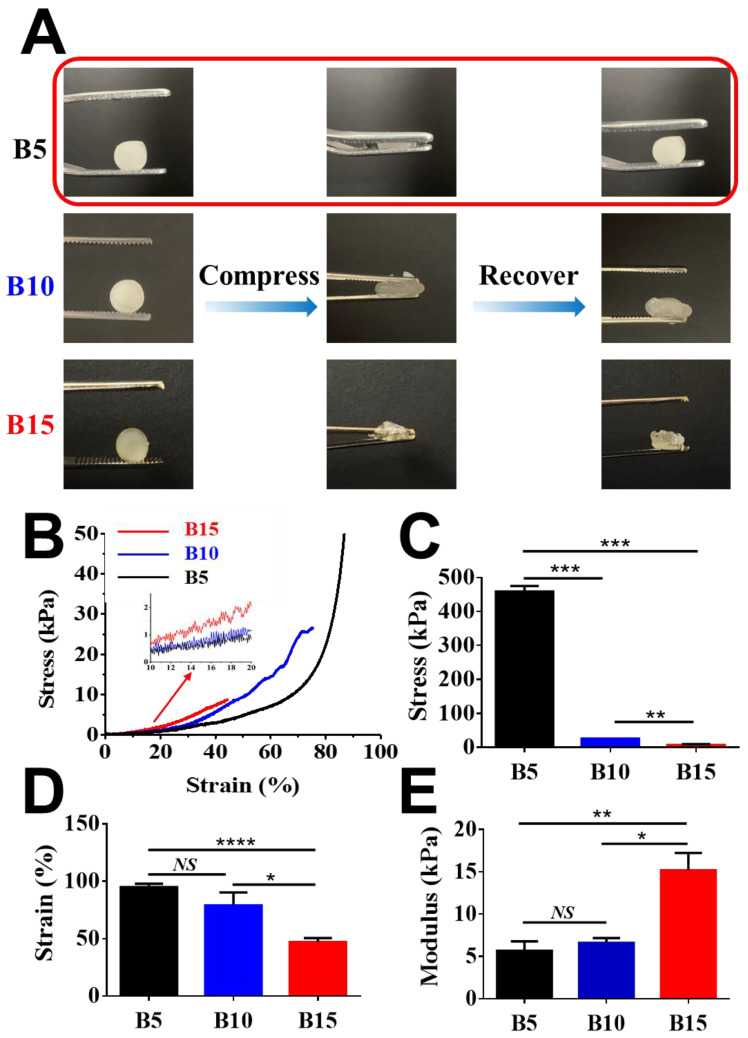
Mechanical properties of BSAMA cryogels (B5, B10, and B15). (**A**) Shape recovery experiments. 5% *w*/*v* BSAMA cryogels (B5) returned to the original shape when the stress was removed. (**B**) Strain–stress curves were recorded during the axial loading. (**C**) Maximum stress at a break point. (**D**) Maximum strain at a break point. (**E**) Young’s modulus was determined by the slope of the stress–strain curve from 10% to 20% strain. *n* = 3, *: *p* < 0.05; **: *p* < 0.01, ***: *p* < 0.005, ****: *p* < 0.001, NS means no significant difference between the samples at the extremities of the line.

**Figure 4 gels-08-00404-f004:**
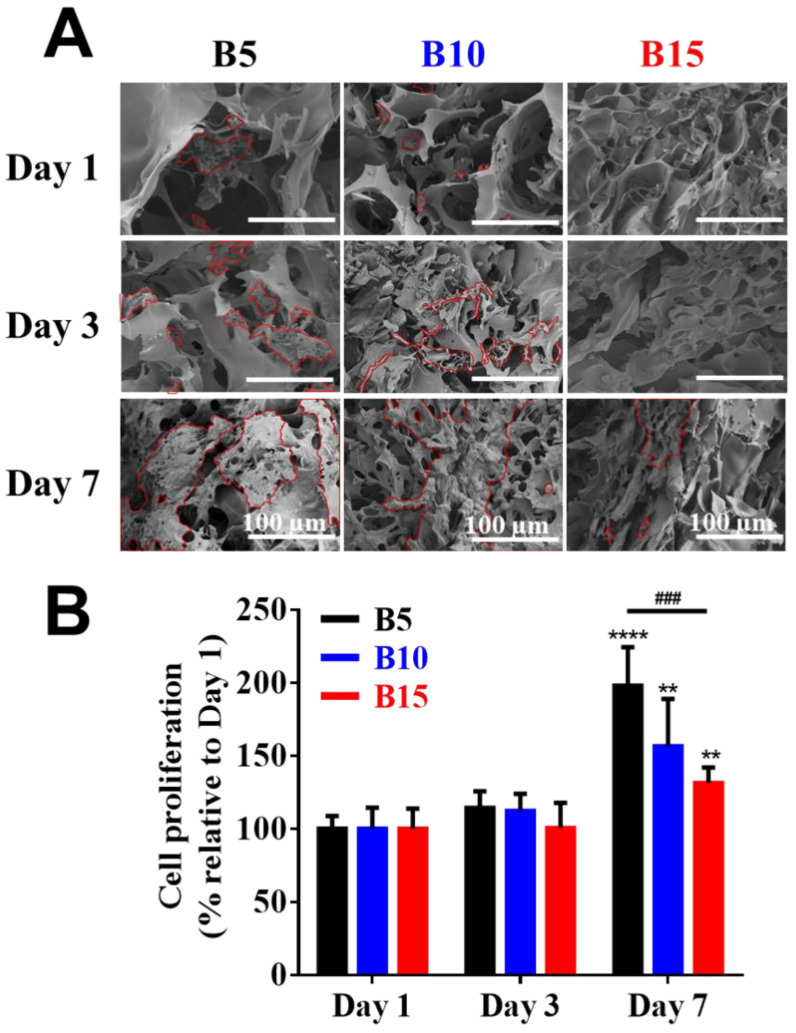
Cell viability on BSAMA cryogels (B5, B10, and B15) for 7 days. (**A**) SEM images of the inside of cell-laden cryogels at day 1, day 3, and day 7. The outline of cells is marked in red. Scale bar: 100 μm. (**B**) Cell proliferation. *n* = 3, **: *p* < 0.01, ****: *p* < 0.001, compared to the same cryogel type on the day 1. ###: *p* < 0.005, between the samples at the extremities of the line.

**Figure 5 gels-08-00404-f005:**
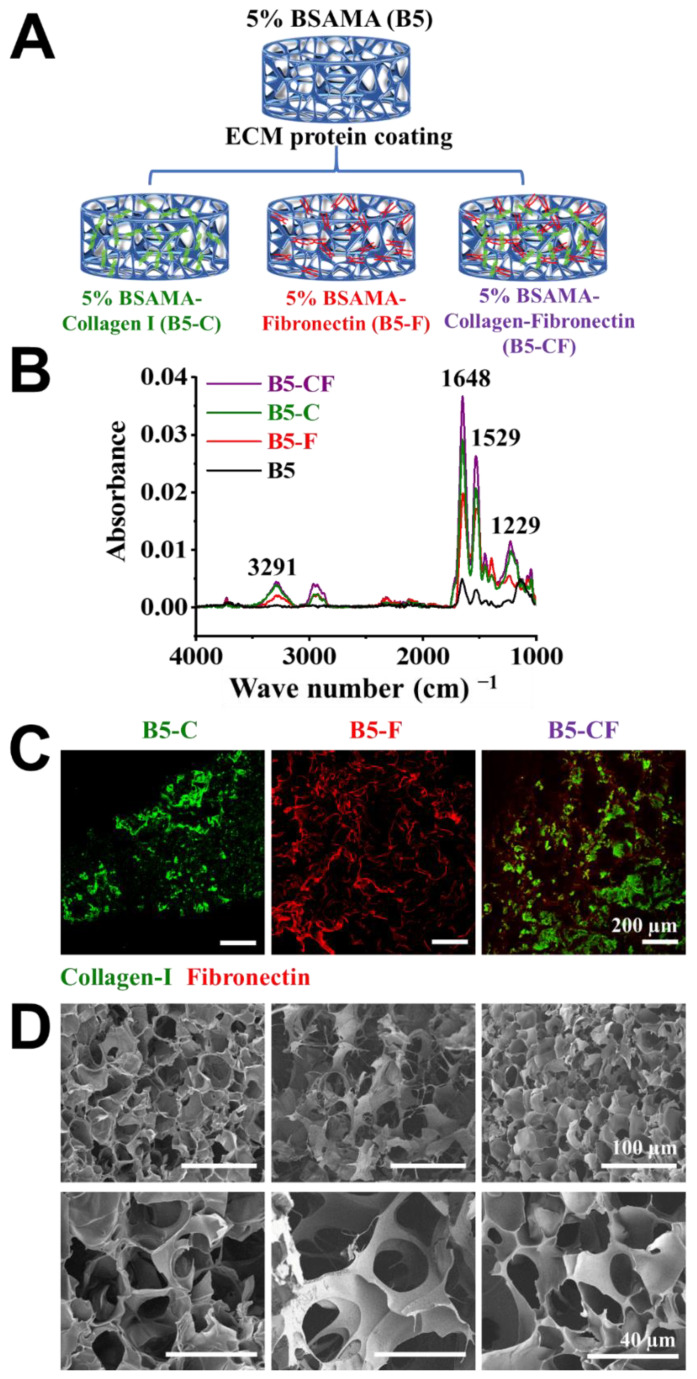
Preparation and characterization of ECM-coated BSAMA cryogels. (**A**) Schematic illustration presenting the preparation of 5% BSAMA–collagen I (B5-C), 5% BSAMA–fibronectin (B5-F), 5% BSAMA–collagen I–fibronectin (B5-CF). (**B**) FTIR spectra of unmodified and ECM protein-modified BSAMA cryogels. The broad peaks at 3118–3490 cm^−1^ resulting from a N-H stretching frequency of amine groups (-NH2). Peaks at 1648 cm^−1^, 1529 cm^−1^, and 1229 cm^−1^ correspond to the stretching/bending of amide-I, amide-II, and amide-III groups, respectively. (**C**) Immunostaining of collagen I and fibronectin in the ECM-coated BSAMA cryogels. Collagen I was stained with DyLight 488 (Green) and fibronectin was stained with DyLight 594 (red). Scale bar: 200 μm. (**D**) SEM images of ECM-modified BSAMA cryogels for ensuring that the overall porous structure of the cryogels appeared intact even after the coating. Scale bar: 100 μm (top); 40 μm (bottom).

**Figure 6 gels-08-00404-f006:**
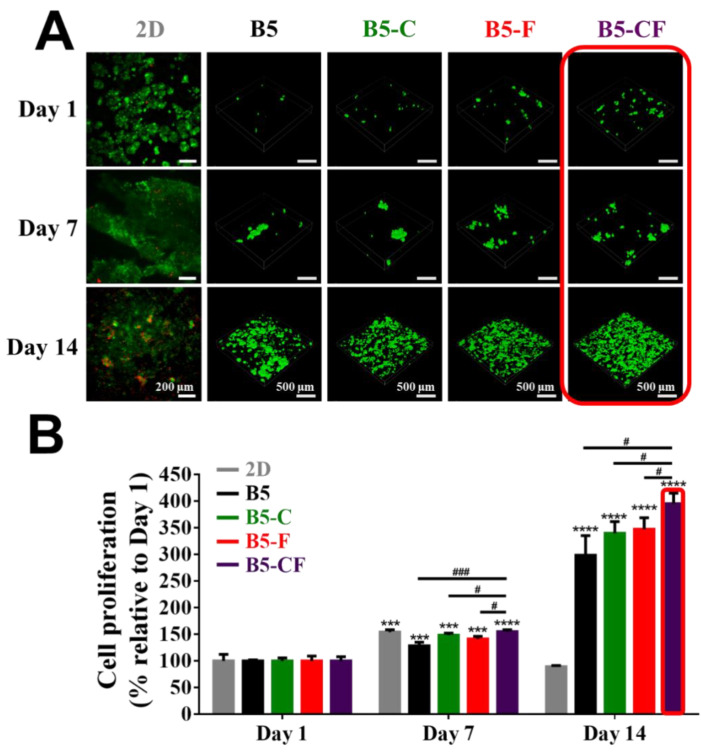
Cell proliferation of 2D, B5, B5-C, B5-F, and B5-CF samples for 14 days. (**A**) Cell viability was qualitatively assessed via the live/dead assay kit at day 1, day 7, and day 14. Live cells were stained green using calcein-AM, and dead cells were stained red using EthD-1. The 3D images of the inside of cell-laden BSAMA cryogels were taken with a confocal microscope with a 10 × lens. Scale bar: 200 μm (2D); 500 μm (3D). (**B**) Cell proliferation was quantitatively detected by CCK-8. *n* = 3, ***: *p* < 0.005, ****: *p* < 0.001, compared to the same cryogel type at day 1. #: *p* < 0.05, ###: *p* < 0.005, between two samples at the extremities of the line.

**Figure 7 gels-08-00404-f007:**
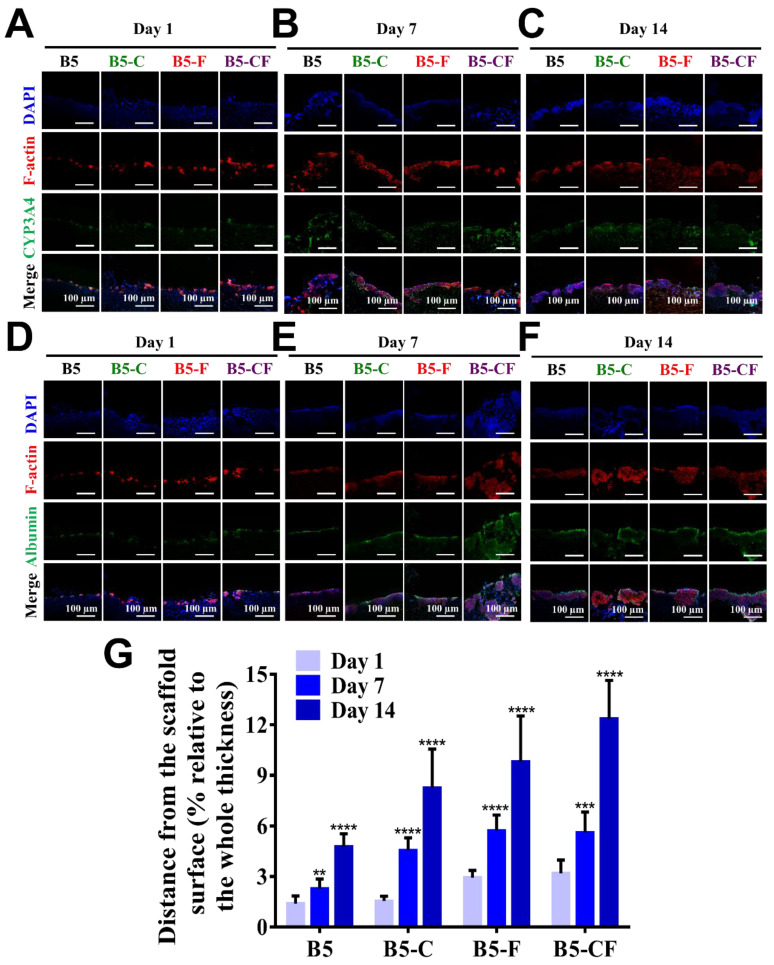
Evaluation of liver-specific functions and cellular infiltration of HepG2 cell constructs in cross-sections of 3D BSAMA cryogels at day 1, day 7, and day 14. (**A**–**C**) Confocal microscopic images of CYP3A4 immunostaining. Blue: nucleus; red: F-actin; green: CYP3A4. (**D**–**F**) Confocal microscopic images of albumin immunostaining. Blue: nucleus; red: F-actin; green: albumin. Scale bar: 100 μm. (**G**) Distance of the cell migration. (*n* = 3, **: *p* < 0.01, ***: *p* < 0.005, ****: *p* < 0.001), compared to the same cryogel type at day 1.

**Figure 8 gels-08-00404-f008:**
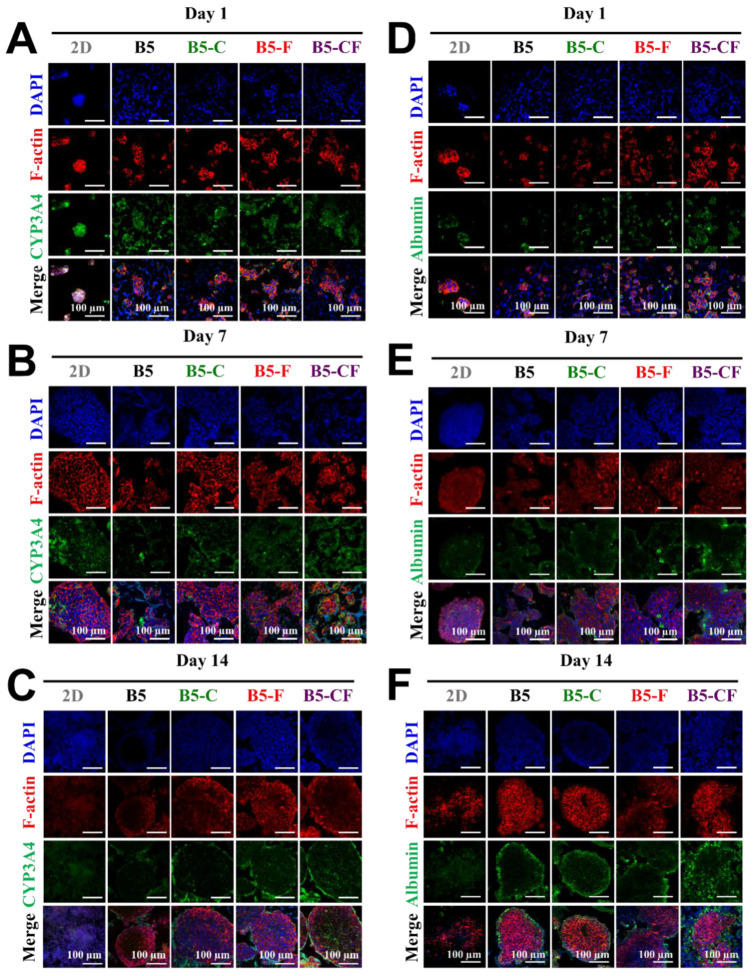
Evaluation of liver-specific functions of HepG2 cell constructs in 3D BSAMA cryogels and on 2D substrates at day 1, day 7, and day 14. (**A**–**C**) Confocal microscopic images of CYP3A4 immunostaining. Blue: nucleus; red: F-actin; green: CYP3A4. (**D**–**F**) Confocal microscopic images of albumin immunostaining. Blue: nucleus; red: F-actin; green: albumin. Cell constructs in B5-CF displayed the highest protein expression of both CYP3A4 and albumin. Scale bar: 100 μm.

**Figure 9 gels-08-00404-f009:**
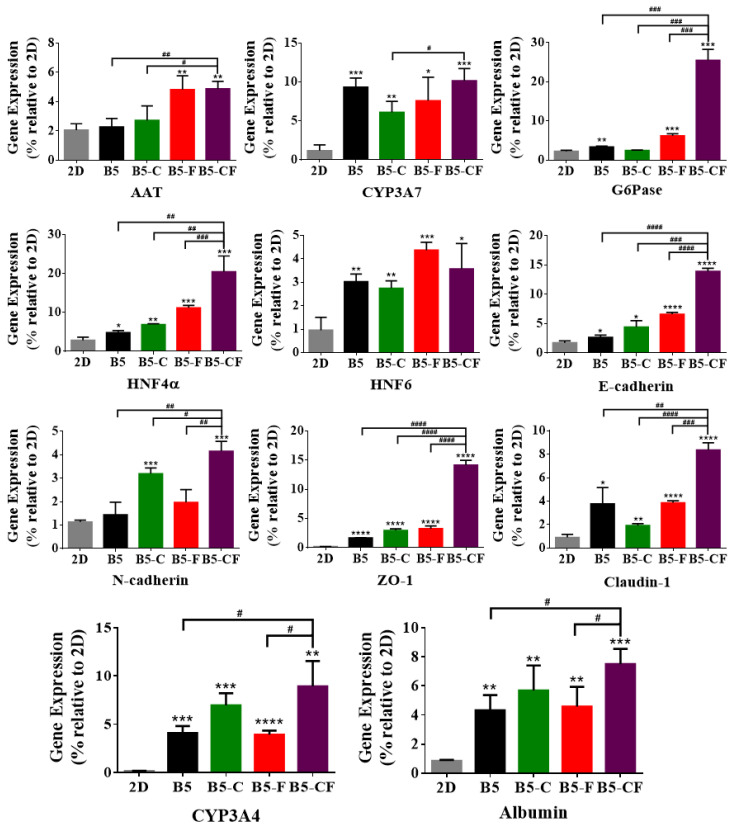
Effect of HepG2 cell culture in 3D BSAMA cryogels and on 2D substrates on liver-specific gene expression at day 14. HepG2 cells were cultured in 3D and 2D culture systems, and their RNA was extracted for the quantitative real-time PCR analysis of AAT, CYP3A7, G6Pase, HNF4α, HNF6, E-cadherin, N-cadherin, ZO-1, claudin-1, CYP3A4, albumin. The data were normalized to the housekeeping gene GAPDH. (*n* = 3, *: *p* < 0.05, **: *p* < 0.01, ***: *p* < 0.005, ****: *p* < 0.001, compared to 2D culture. #: *p* < 0.05, ##: *p* < 0.01, ###: *p* < 0.005, ####: *p* < 0.001, between the samples at the extremities of the line.) AAT: alpha 1-antitrypsin, G6Pase: glucose 6-phosphatase, HNF: hepatocyte nuclear factors, ZO-1: zonula occludens.

**Figure 10 gels-08-00404-f010:**
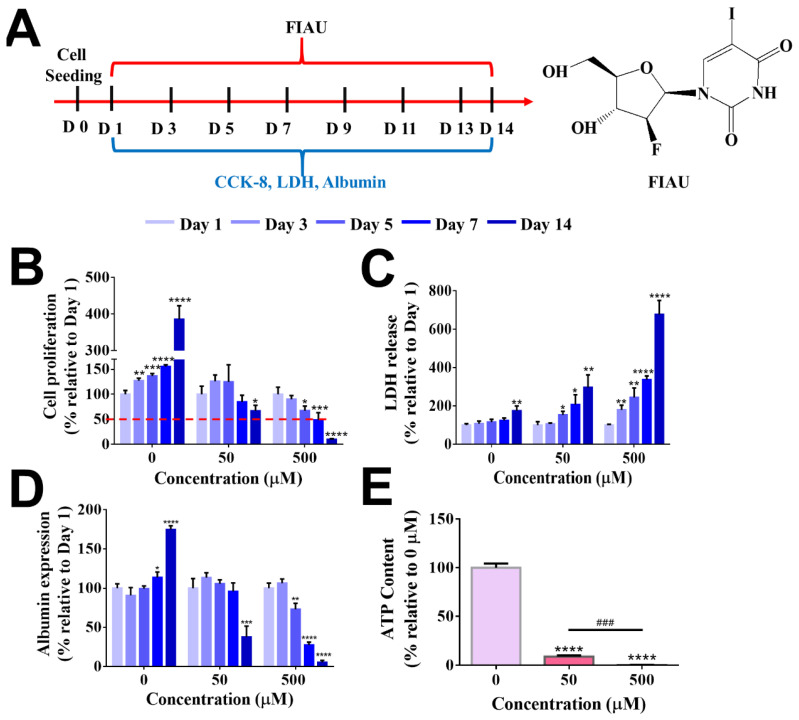
Evaluation of FIAU-induced cytotoxicity in B5-CF scaffolds at day 1, day 3, day 5, day 7, and day 14. (**A**) Experimental timeline for the drug administration. (**B**) Cell proliferation. (**C**) Cytotoxicity (LDH release). (**D**) Albumin expression was assessed at different time points after FIAU treatment (*n* = 3, *: *p* < 0.05, **: *p* < 0.01, ***: *p* < 0.005, ****: *p* < 0.001), compared to the same cryogel type at day 1. (**E**) Evaluation of FIAU-induced human-specific toxicity, mitochondrial disfunction in B5-CF scaffolds. ATP content in cell-laden B5-CF was assessed at the end of 14 days after FIAU treatment. *n* = 3, ***: *p* < 0.005, ****: *p* < 0.001, compared to the group without FIAU treatment. ###: *p* < 0.005, between the samples at the extremities of the line. The red dashed line indicates the 50% relative cell proliferation.

## Data Availability

The data generated from the study are clearly presented and discussed in the manuscript.

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
