# Peer review of "An Engineered Protein-Based Building Block (Albumin Methacryloyl) for Fabrication of a 3D In Vitro Cryogel Model"

_gels, 2022, doi:10.3390/gels8070404_

Round 1
Reviewer 1 Report
Xueming Niu et al reported albumin-based cryogel in
vitro model provided improved cell-cell and cell-material interactions and consequently displayed excellent liver functional gene expression, being conducive to detection of fialuridine (FIAU) hepa- totoxicity . I will consider publishing your current review paper entitled “An engineered protein-based building block (albumin methac- 2
ryloyl) for fabrication of a 3D in vitro cryogel model ” after minor revision. Although the subject of this manuscript is very interesting, the content a little disappointing, and doesn´t fit neither the Title nor the Abstract.
1. I had a close look and found the manuscript is partially hard to read and comprehend, but the manuscript should be rewritten more readable.
2. The authors need to compare their results with published articles.
3. The authors should provide some information or discussion on the composition or chemical structure of the used materials.
4. The Conclusion should provide a critical (!) assessment in comparison with the related papers, including limitations and advantages.
5. How the authors estimated the exact size of the synthesized nanoparticles? Image J!!! it is better to use BET analysis.
6. some references should be added. Some of the related references are given as well:
https://doi.org/10.1016/j.cis.2020.102316
Author Response
Reviewer #1:
Xueming Niu et al reported albumin-based cryogel in vitro model provided improved cell-cell and cell-material interactions and consequently displayed excellent liver functional gene expression, being conducive to detection of fialuridine (FIAU) hepa- totoxicity . I will consider publishing your current review paper entitled “An engineered protein-based building block (albumin methacryloyl) for fabrication of a 3D in vitro cryogel model” after minor revision. Although the subject of this manuscript is very interesting, the content a little disappointing, and doesn´t fit neither the Title nor the Abstract.
Response: Thank you much for your comments. We have revised the abstract. In our work, for the first time we propose the fabrication of an albumin methacryloyl cryogel platform inspired by the liver’s microarchitecture via emulating the mechanical properties and extracellular matrix (ECM) cues of liver. Engineered crosslinkable albumin methacryloyl is used as a protein-based building block for fabrication of albumin cryogel in vitro models that can have potential applications in 3D cell culture and drug screening.
Q1. I had a close look and found the manuscript is partially hard to read and comprehend, but the manuscript should be rewritten more readable.
Response: Thank you for your comments. We have polished our writing and double-check each sentence throughout the manuscript.
Q2. The authors need to compare their results with published articles.
Response: Thank you much for your comments. According to the reviewer’s suggestion, we have further discussed our results in comparison to published articles. We have highlighted further discussion in yellow in the revised manuscript.
Q3. The authors should provide some information or discussion on the composition or chemical structure of the used materials.
Response: Thank you much for your suggestion. We have further provided some information/discussion on the chemical structure of the used materials (e.g. BSAMA and FIAU) in experiments and methods as well as results and discussion. We have highlighted them in yellow.
Q4. The Conclusion should provide a critical (!) assessment in comparison with the related papers, including limitations and advantages.
Response: Thank you much for your excellent comment. We have further mentioned limitations and future work of our albumin-based cryogel system in our conclusion.
Q5. How the authors estimated the exact size of the synthesized nanoparticles? Image J!!! it is better to use BET analysis.
Response: Thank you for your comment. We used Image J to measure the pore size inside our cryogels.
Q6. some references should be added. Some of the related references are given as well:
https://doi.org/10.1016/j.cis.2020.102316
Response: Thank you for this suggestion. We have added additional related references in our revised manuscript.
Reviewer 2 Report
The manuscript describes the development of a 3-D scaffold based on cross-linked albumin to mimic the liver microenvironment. The paper reads very well, with few minor typos. Presents a consistent study from materials preparation to the assessment of the scaffold (cryogel) to assess the hepatotoxicity of FIAU using HepG2 (immortalized) cells. I would suggest that a zoom of the stress-strain curve (Figure 3B) in the region 10-20% strain should be added to the Supplementary Material
Reviewer 3 Report
In this paper, the authors have investigated the production and characterization of albumin methacryloyl-based 3D cryogels having high interconnectivity, liver-like stiffness as in vitro models for 3D culture and drug screening. Overall, the paper is well-structured. Conclusions are supported by data and clear. Below a couple of minor comments.
LINES 138-140: How have the samples for SEM analyses been prepared? I guess the authors have removed the water entrapped within the gel-based matrix. Please add a comment on this aspect.
LINES 148-149: I would add a comment on the fact that all the experiments have been carried out on constant freezing conditions. Further control of the average pore size of ice crystals might be obtained by controlling the freezing conditions, e.g., cooling rate during cryogelation. I would add a comment on that and appropriate ref. (DOI:10.1080/07373937.2018.1528451).
